# Efficacy and Safety of Heterologous Booster Vaccination after Ad5-nCoV (CanSino Biologics) Vaccine: A Preliminary Descriptive Study

**DOI:** 10.3390/vaccines10030400

**Published:** 2022-03-05

**Authors:** José Francisco Muñoz-Valle, Gabriela Athziri Sánchez-Zuno, Mónica Guadalupe Matuz-Flores, Cristian Oswaldo Hernández-Ramírez, Saúl Alberto Díaz-Pérez, Christian Johana Baños-Hernández, Francisco Javier Turrubiates-Hernández, Alejandra Natali Vega-Magaña, Jorge Hernández-Bello

**Affiliations:** Institute of Research in Biomedical Sciences, University Center of Health Sciences (CUCS), University of Guadalajara, Guadalajara 44340, Mexico; drjosefranciscomv@cucs.udg.mx (J.F.M.-V.); athziri.sanchez@alumnos.udg.mx (G.A.S.-Z.); monica.matuz3435@alumnos.udg.mx (M.G.M.-F.); cristian.hernandez3703@alumnos.udg.mx (C.O.H.-R.); saul.diaz2330@alumnos.udg.mx (S.A.D.-P.); johana.banos@academicos.udg.mx (C.J.B.-H.); francisco.turrubiates3337@alumnos.udg.mx (F.J.T.-H.); alejandra.vega@academicos.udg.mx (A.N.V.-M.)

**Keywords:** COVID-19, SARS-CoV-2, antibodies, vaccination, booster

## Abstract

Several studies have reported the benefits and safety of heterologous vaccination among different approved vaccines; however, there are no specific reports on the effects of vaccination with the Ad5-nCoV and other vaccines of the same or different technologies. In the present study, we evaluated the neutralizing antibodies percentage against SARS-CoV-2 in Mexican patients immunized with the Ad5-nCoV vaccine six months after its application. Moreover, the effect of the heterologous vaccination with the Ad5-nCoV vaccine and a booster dose of ChAdOx1-S-Nov-19, Ad26.COV2.S, BNT162b2, or mRNA-127 were determined. Our results suggest that a heterologous regimen of one dose with Ad5-nCoV vaccine followed by a booster dose of a different vaccine is safe and induces a stronger humoral immune response.

## 1. Introduction

The development of vaccines against COVID-19 is a preventive public health strategy aimed at avoiding cases of infection and reducing the mortality rate associated with the disease [1,2]. The rapid technological progress to affront the COVID-19 pandemic has accelerated the development of several vaccine types, including viral vector-based vaccines, mRNA and DNA vaccines, subunit vaccines, nanoparticle-based vaccines, and inactivated-whole virus vaccines [3].

The COVID-19 pandemic has forced health authorities around the world to design different vaccination plans; in Mexico, it was decided to start the vaccination in a population considered at risk, including health and educational personnel in which the latter group was exclusively administered the Ad5-nCoV vaccine [4,5]. The Ad5-nCoV vaccine (developed by the Beijing Institute of Biotechnology, Beijing, China, and CanSino Biologics, Tianjin, China) has only been approved in countries such as Argentina, Chile, China, Ecuador, Hungary, Indonesia, Malaysia, Mexico, Pakistan, and Republic of Moldova [6,7]. This vaccine contains replication-defective Ad5 vectors expressing the full-length spike gene based on Wuhan-Hu-1. The phase I and II trials results suggest that this vaccine demonstrated a good safety profile [8,9,10]. Despite this information, the lack of authorization by the WHO for the Ad5-nCov vaccine generates uncertainty in the population; thus, it is important to gain more knowledge about the immunogenicity, efficacy, duration of protection, and side effects to contribute to an informed decision-making process for the health authorities.

In a previous study, we evaluated the levels of neutralizing antibodies against SARS-CoV-2 in Mexican individuals immunized with the Ad5-nCoV vaccine. One of the main findings was that 7.4% of the individuals did not present positive titers of neutralizing antibodies after vaccination. Moreover, neutralizing percentages were lower than those observed in the first dose of the BNT162b2 vaccine (Pfizer-BioNTech); therefore, the application of a second booster dose was suggested, especially in older age and immunosuppressed patients [11]. On this basis, one of the questions posed by the scientific community and the Mexican population is: Is the application of combined vaccines recommended?

Some studies have reported the benefits and security of heterologous vaccination between different approved vaccines [12,13,14]. Regarding the Ad5-nCoV vaccine, it was recently reported that individuals immunized with one dose of Ad5-nCoV and a booster dose of the BNT162b2 vaccine presented higher titers of IgG antibodies against SARS-CoV-2 compared to individuals who received only one dose of Ad5-nCoV [15]. However, only the presence of IgG antibodies was evaluated and not their neutralizing capacity. Therefore, this study aimed to determine the effect of the heterologous vaccination of the Ad5-nCoV vaccine with ChAdOx1-S-nCoV-19 (AstraZeneca), BNT162b2 (Pfizer-BioNTech), mRNA-127 (Moderna), or the Janssen COVID-19 vaccine.

## 2. Materials and Methods

### 2.1. Subjects

From 28 April to 4 May 2021, the Mexican government performed a vaccination campaign against COVID-19 with the Ad5-nCoV vaccine for teachers from Jalisco, Mexico. On the last week of May 2021 (21–24 days after vaccination), some of those professors were invited to voluntarily participate in a study to evaluate reactogenicity to the Ad5-nCoV vaccine, the production of neutralizing antibodies against SARS-CoV-2, and the anti-Ad5 Antibodies in response to this vaccine [11]. Six months later, the previous study participants were again invited to assess a possible decline in neutralizing titers. Sixty-two people declared that they voluntarily received a heterologous booster 4.5–5 months after applying the Ad5-nCoV vaccine. Those 62 professors and 62 individuals without a booster dose were included in the present study. Both groups were matched for age, gender, treatment, COVID-19 history, and baseline antibody levels 21 days after the first shot.

Two surveys were applied to all participants (N = 124) to obtain clinical and demographic data, history of SARS-CoV-2 infection, and vaccine-associated side effects from the Ad5-nCoV as a single shot or a heterologous booster. The first survey was applied 21–24 days after the first shot and the other one six months after that vaccine application. Additionally, a blood sample was obtained from both study groups to analyze neutralizing antibodies against SARS-CoV-2 in response to vaccination.

Individuals with prior COVID-19 were diagnosed 1–12 months before the study by RT-PCR (reverse transcription-polymerase chain reaction).

### 2.2. Quantification of Neutralizing Antibodies

The quantification of neutralizing antibodies was performed in the serum collected after six months of the immunization with the Ad5-nCoV vaccine. The cPass™ SARSCoV-2 Neutralization Antibody Detection Kit (GenScript, Piscataway Township, NJ, USA; Cat. L00847-A Kit) was used for this analysis, which is a blocking Enzyme-Linked Immunosorbent Assay (ELISA) designed to detect immunity against the original Wuhan strain of SARS-CoV-2.

This kit is validated for diagnosis with a 30% signal inhibition cut-off point for SARS-CoV-2 neutralizing antibody detection. According to the manufacturer’s instructions, negative and positive sample controls were diluted at 1:10 with the sample dilution buffer and mixed with an equal volume of HRP-conjugated RBD (60 μL and 60 μL). They were then incubated at 37 °C for 30 min. Then, 100 μL of this mixture was transferred to 96-well plates coated with recombinant hACE2 and incubated at 37 °C for 15 min. After the incubation, the supernatant was removed, and the plates were washed four times with the wash solution. Finally, 100 μL of tetramethylbenzidine (TMB) was added and incubated for 15 min at room temperature. The reaction was stopped with 50 μL of stop solution, and plates were read at 450 nm immediately after. The inhibition rate was calculated as follows:$$\%\;\mathrm{signal}\;\mathrm{inhibition}=\left(1-\frac{\mathrm{OD}\;\mathrm{value}\;\mathrm{sample}}{\mathrm{OD}\;\mathrm{value}\;\mathrm{of}\;\mathrm{Negative}\;\mathrm{Control}}\right)\times 100$$

### 2.3. Statistical Analysis

Statistical analysis was performed using the GraphPad Prism v. 6.01 software (GraphPad company, San Diego, CA, USA) and the R version 4.0.0 statistical software program (R core Team, Vienna, Austria). The significance level was set at *p* < 0.05. For simple comparisons, we used the Fisher exact test or Student’s *t*-test. Data with nonparametric distribution were represented as median with interquartile range (IQR). The Mann–Whitney U-test was applied for comparing two groups and the Kruskal–Wallis test was applied for three or more group comparisons.

## 3. Results

### 3.1. Clinical and Demographic Characteristics

The clinical and demographic characteristics of both study groups are described in Table 1. A total of 124 individuals vaccinated with the Ad5-nCoV vaccine were considered for the analysis, of which 62 received a booster dose with one of the following vaccines: AstraZeneca, Moderna, Pfizer, or Johnson & Johnson. The age and gender of the individuals were similar in both study groups.

Both groups were similar in comorbidities and treatments (*p* > 0.05). Moreover, no differences were found for clinical and demographic characteristics when stratifying by different booster vaccines (Table 2).

In the baseline determination carried out 21 days after the individuals received the Ad5-nCoV vaccine, the median percentage of neutralizing antibodies was similar in both groups (78.16% vs. 78.65%) (*p* > 0.05). However, six months later, the median percentage of neutralizing antibodies was significantly higher in individuals who received a booster dose (96.41% vs. 89.33%, *p* = 0.0004) (Figure 1a). The 62 individuals with a booster were grouped according to the vaccine applied to assess whether any combination showed a greater capacity to generate neutralizing antibodies; however, no differences were observed in this regard (Figure 1b).

The neutralization percentage did not correlate with the time elapsed between the first vaccine and the booster (data not shown, r = −0.06, *p* = 0.62),

### 3.2. Adverse Effects in Individuals with or without Booster Doses

Table 3 shows the adverse effects (reactogenicity) of the Ad5-nCoV vaccine with or without a booster dose with a different vaccine. The presence of myalgia and fatigue was more frequent in individuals after receiving the Ad5-nCoV vaccine than in those who received a booster dose with a different vaccine (*p* = 0.0441 and *p* < 0.000, respectively). No differences were found regarding adverse effects when comparing the different booster vaccines (Table 4).

## 4. Discussion

Progress in the development and emergency approval for vaccines against the SARS-CoV-2 virus is being fast-tracked globally, mainly due to the need to control the COVID-19 lethality. The shortage of COVID-19 vaccines and the low purchasing power of developing countries have led some countries to approve vaccines that have not completed phase III clinical trials at the time of their application, such as the Ad5-nCoV vaccine (commercial name “Convidicea”), which was approved for emergency use in the populations of Argentina, Chile, China, Ecuador, Hungary, Indonesia, Malaysia, Pakistan, Republic of Moldova, and Mexico [7].

The phase III study of this vaccine has just been reported; 28 days after vaccination, efficacy against PCR-confirmed COVID-19 was found to be 57.5% and 91.7% protective against severe COVID-19. Moreover, seroconversion of neutralizing antibodies was shown in 75.9% of Ad5-nCoV recipients. It is also noteworthy that efficacy was substantially lower (17.5%) in participants aged ≥ 60 years than participants <60 years, suggesting that additional vaccine doses might be necessary for this age group [16]. On the other hand, that phase III study presents some limitations, such as the short follow-up time (<2 months for most people) and that individuals with compromised immune systems, unstable medical conditions, and other potential risks were excluded. Therefore, it is necessary to wait until real-world effectiveness studies are carried out to ascertain the ability of the vaccine to protect these vulnerable groups. Moreover, the short follow-up means that the current report cannot provide additional information to address current concerns about the longevity of vaccine-induced protection [17].

Our research group previously carried out the first study outside the clinical trials to evaluate the capacity of the Ad5-nCoV vaccine to induce the production of neutralizing antibodies as well as the side effects associated with it. In the present study, we compare the decline of neutralizing antibodies’ percentage against SARS-CoV-2 six-months after immunization with the Ad5-nCoV vaccine in individuals with a single shot or with a heterologous vaccination regimen (Ad5-nCoV and a booster dose of ChAdOx1-S-nCoV-19, Ad26.COV2.S, BNT162b2, or mRNA-127).

Our results indicated that recipients of the heterologous booster regimen had higher percentages of antibodies with neutralizing capacity than those who only received the Ad5-nCoV vaccine. These results are consistent with previous studies where the effects of other heterologous vaccination regimens were evaluated [12,13,14,15,18,19,20]. In fact, according to these reports, the effectiveness of a heterologous vaccination scheme was significantly higher compared to homologous schemes with ChAdOx1 nCoV-19 or BNT162b2 [12,14,18,19,20].

The mRNA technologies have proven to be excellent candidates to be combined with vaccines of different technology and thus broaden the spectrum of protection against SARS-CoV-2 variants. A combined regimen can offer complementary stimulation to different immune pathways and thus induce more effective humoral and cellular responses [21].

Heterologous prime-boost doses of COVID-19 vaccines are being studied in two principal clinical trials (‘Com-Cov2 trial’ in the UK and ‘CombivacS trial’ in Spain) [22,23]. Together, both trials include the analysis of the effectiveness of combinations with Oxford (AZD1222), Pfizer (BNT162b2), Moderna (mRNA-1273), and Novavax (NVX-CoV2373) vaccines for COVID-19 [22].

One of the major limitations of the COVID-19 viral vector technologies such as the Ad5-nCoV vaccine is the likely presence of pre-existing antibodies against strains of adenovirus. Zhu et al. reported that the Ad5-vectored vaccine might have lower immunogenicity in populations with a high prevalence of pre-existing anti-Ad5 immunity [6]. The ChAdOx1-S-nCoV-19 vaccine partly overcomes this limitation since the latter was designed with a chimpanzee adenovirus to which there is no common exposure [24].

In Mexico, there are no reports on the seroprevalence of anti-Ad5 antibodies in the general population; however, a study reported that around 23% of cases associated with acute respiratory infections in children were related to adenovirus [25]. In addition to this, in our previous study [11], we reported a slight increase in the presence of anti-Ad5 antibodies after vaccination with Ad5-now. All this suggests that a detailed analysis should be carried out to examine whether it is convenient to use a booster dose with the same vaccine for those who have received the Ade5-nCoV vaccine or whether, the heterologous vaccination scheme, using ChAdOx1 nCoV-19, for example could be most beneficial.

In the present study, which supported the idea for the safety of a heterologous vaccination for Ade5-nCoV, it was not observed that the application of a booster with a heterologous vaccine had aroused greater side effects when compared to those reported in the first/single dose. The frequency in myalgia and fatigue was significantly lower (*p* < 0.005) in individuals who received a booster dose. These results may be directly related to the fact that the initial dose of the Ad5-nCoV vaccine represents a “primer” for the immune system, this first encounter triggers the production of antibodies as well as a cellular response that is directly associated with side effects such as fatigue and myalgia [23].

Most of the patients included in the present study received the booster dose 4.5–5 months after the first Ade5-nCoV dose, so this timing could be a starting point to consider as an adequate time for a booster. The neutralization percentage did not correlate with the time elapsed between the first vaccine and the booster, nor was there a difference between the neutralization percentage induced by each of the vaccines used as a booster. Similarly, we did not observe differences in the side effects associated with each booster shot. Therefore, our results suggest a booster heterologous regimen (Ad5-nCoV vaccine followed by ChAdOx1-S-nCoV-19, Ad26.COV2.S, BNT162b2, or mRNA-127z) could be safe and result in a robust humoral immune response. We only observed one patient who, after booster with the Moderna vaccine, did not generate neutralizing antibodies (<30% neutralization). This was a patient with a history of autoimmunity and the use of immunosuppressants who did not generate neutralizing antibodies since the first dose.

The arrival of new variants, such as Delta and Omicron, demands an urgent vaccination strategy that allows the immunization and protection provided by vaccination to be optimized, specifically in those vulnerable groups and those in which the efficacy of the vaccine has not yet been fully proven as for Ade5-nCov vaccine.

Data from Israel and the United Kingdom indicate that a booster dose of one of the widely used mRNA-based vaccines sharply lowers a person’s likelihood in catching SARS-CoV-2 and getting sick [26]. Heterologous prime-boost trials have shown safety, effectiveness, good tolerability with improved immunogenicity, and flexibility profiles for future vaccinations, especially during acute and global shortages, compared to the homologous counterparts [22].

In conclusion, our preliminary data support heterologous booster dose benefits and their possible safety in those previously vaccinated with Ad5-now. One of the most important limitations of the present study is the small sample size used, which is why larger-scale studies are needed to verify our findings. Therefore, the observed findings must be confirmed in cohorts with a larger number of individuals and interpreted with caution.

Likewise, there is no access to individuals vaccinated with two doses of Ade5-nCoV to compare the response between this scheme and heterologous vaccination because the Ade5-nCoV vaccine was designed as a single shot. In addition, future studies will be needed to determine the long-term effectiveness of the booster dose against current and new emerging variants.

## Figures and Tables

**Figure 1 vaccines-10-00400-f001:**
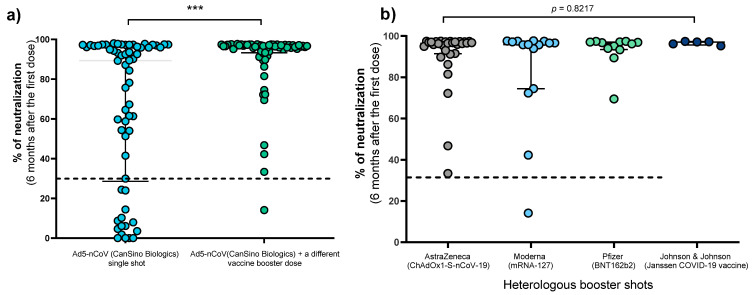
Comparison of antibody neutralizing signals between individuals with a single dose of Ad5-nCoV vaccine and individuals with a heterologous booster dose. (**a**) Individuals with a single dose of Ad5-nCoV vaccine vs. individuals with Ad5-nCoV dose and heterologous booster doses (AstraZeneca, Moderna, Pfizer, or Johnson & Johnson), differences were calculated by the Mann—Whitney’s U test. (**b**) Schemes of individuals with heterologous booster doses (Ad5-nCoV and AstraZeneca, Moderna, Pfizer, or Johnson & Johnson), differences were calculated by the Kruskal–Wallis test. The dotted line indicates the cut-off point for the neutralization test (>30%). ***, *p* < 0.0001.

**Table 1 vaccines-10-00400-t001:** Clinical and demographic characteristics of the study groups.

	Immunized with Ad5-nCoV Vaccine		*p*-Value
	With a Different Vaccine Booster *n* = 62	Without any Booster *n* = 62	
Age (years), ^mean ± SD^	41.00 ± 9.96	41.34 ± 10.55	0.854
Gender, *^n^* ^(%)^			
Female	47 (75.81)	43 (69.35)	0.420
Male	15 (24.19)	19 (30.65)	
COVID-19 prior to vaccination, *^n^* ^(%)^	19 (30.65)	19 (30.65)	1.000
Basal neutralizing antibodies (%), ^median (IQR)^	78.16 (50.31–97.72)	78.65 (65.94–97.67)	0.303
Comorbidities, ^*n* (%)^			
None	28 (45.16)	26 (41.94)	0.717
Overweight/obesity	18 (29.03)	22 (35.48)	0.442
Allergic diseases	8 (12.90)	9 (14.52)	0.794
SAH	3 (4.84)	5 (8.06)	0.465
Diabetes	1 (1.61)	4 (6.45)	0.364
Hypothyroidism	3 (4.84)	1 (1.61)	0.618
Autoimmune diseases	4 (6.45)	0 (0.00)	0.119
Dermatitis	2 (3.23)	1 (1.61)	1.000
Dyslipidemia	2 (3.23)	1 (1.61)	1.000
Heart diseases	0 (0.00)	2 (3.23)	0.495
Treatment, *^n^* ^(%)^			
Antidepressants	10 (16.13)	5 (8.06)	0.455
Antihypertensive	4 (6.45)	5 (8.06)	0.729
Hormonal	6 (9.68)	2 (3.23)	0.272
Hypoglycemic agents	2 (3.23)	3 (4.84)	1.000
Hypolipidemic agents	1 (1.61)	1 (1.61)	1.000
NSAIDs	1 (1.61)	1 (1.61)	1.000
Antihistamines	2 (3.23)	0 (0.00)	0.495
Immunosuppressants	1 (1.61)	0 (0.00)	1.000

IQR: interquartile range; SAH: systemic arterial hypertension; NSAIDs: non-steroidal anti-inflammatory drugs; SD: standard deviation.

**Table 2 vaccines-10-00400-t002:** Clinical and demographic characteristics of individuals with a booster.

	Immunized with Ad5-nCoV Vaccine				*p*-Value
	AstraZeneca (ChAdOx1-S-nCoV-19) *n* = 30	Moderna (mRNA-127) *n* = 15	Pfizer (BNT162b2) *n* = 12	Johnson & Johnson (Janssen COVID-19 Vaccine) *n* = 5	
Age (years), ^mean ± SD^	42.4 ± 10.1	37.8 ± 9.5	42.3 ± 10.6	38.8 ± 8.5	0.462
Gender, *^n^* ^(%)^					
Female	24 (80.00)	13 (86.67)	6 (50.00)	4 (80.00)	0.132
Male	6 (20.00)	2 (13.33)	6 (50.00)	1 (20.00)	
COVID-19 prior to vaccination, *^n^* ^(%)^	11 (36.67)	3 (20.00)	3 (25.00)	2 (40.00)	0.679
Basal neutralizing antibodies, (%), ^median (IQR)^	80.66 (51.95–97.95)	71.22 (55.77–93.73)	67.03 (39.96–89.06)	88.95 (84.51–98.07)	0.288
Days elapsed between the first and last vaccination, ^mean ± sd^	155.2 ± 31.05	147.6 ± 47.07	134.3 ± 31.80	132.2 ± 86.63	0.413
Comorbidities, *^n^* ^(%)^					
None	15 (50.00)	3 (25.00)	6 (50.00)	4 (80.00)	0.079
Overweight/obesity	7 (23.33)	7 (46.67)	4 (33.33)	0 (0.00)	0.189
Allergic diseases	4 (13.33)	2 (13.33)	1 (8.33)	1 (10.00)	0.890
Autoimmune diseases	1 (3.33)	2 (13.33)	1 (8.33)	0 (0.00)	0.581
SAH	2 (6.67)	1 (6.67)	0 (0.00)	0 (0.00)	1.000
Hypothyroidism	2 (6.67)	1 (6.67)	0 (0.00)	0 (0.00)	1.000
Dyslipidemia	1 (3.33)	0 (0.00)	1 (8.33)	0 (0.00)	0.532
Dermatitis	1 (3.33)	1 (6.67)	0 (0.00)	0 (0.00)	1.000
Diabetes	1 (3.33)	0 (0.00)	0 (0.00)	0 (0.00)	1.000
Treatment, *^n^* ^(%)^					
At least one	18 (60.00)	7 (46.67)	5 (41.67)	0 (0.00)	0.091
Antidepressants	6 (20.00)	0 (0.00)	4 (33.33)	0 (0.00)	0.069
Hormonal	3 (10.00)	3 (20.00)	0 (0.00)	0 (0.00)	0.394
Antihypertensive	3 (10.00)	1 (6.67)	0 (0.00)	0 (0.00)	0.859
Antihistamines	0 (0.00)	2 (13.33)	0 (0.00)	0 (0.00)	0.167
Hypoglycemic agents	1 (3.33)	0 (0.00)	1 (8.33)	0 (0.00)	0.532
Hypolipidemic agents	0 (0.00)	0 (0.00)	1 (8.33)	0 (0.00)	0.274
NSAIDs	0 (0.00)	1 (6.67)	0 (0.00)	0 (0.00)	0.516
Immunosuppressants	1 (3.33)	0 (0.00)	0 (0.00)	0 (0.00)	1.000

IQR: interquartile range; SAH: systemic arterial hypertension; NSAIDs: non-steroidal anti-inflammatory drugs.

**Table 3 vaccines-10-00400-t003:** Associated side effects comparison with booster presence.

Side Effects	Immunized with Ad5-nCoV Vaccine		*p*-Value
	With a Different Vaccine Booster *n* = 62 *n* (%)	Without any Booster *n* = 62 *n* (%)	
At least one	34 (54.84)	41 (66.13)	0.1985
Headache	22 (35.48)	27 (43.55)	0.3584
Shivers	20 (32.26)	16 (25.81)	0.4287
Myalgia	12 (19.35)	22 (35.48)	0.0441
Fever	17 (27.42)	13 (20.97)	0.4016
Arthralgia	9 (14.52)	9 (14.52)	1.0000
Irritability	9 (14.52)	9 (14.52)	1.0000
Fatigue	1 (1.61)	24 (38.71)	<0.0000
Application site-pain	4 (6.45)	0 (0.00)	0.1189

**Table 4 vaccines-10-00400-t004:** Booster vaccine-associated side effects comparison.

	Immunized with Ad5-nCoV Vaccine				*p*-Value
	AstraZeneca (ChAdOx1-S-nCoV-19) *n* = 30 *n* (%)	Moderna (mRNA-127) *n* = 15 *n* (%)	Pfizer (BNT162b2) *n* = 12 *n* (%)	Johnson & Johnson (Janssen COVID-19 Vaccine) *n* = 5 *n* (%)	
At least one	17 (56.67)	9 (60.00)	5 (41.67)	3 (60.00)	0.810
Headache	10 (33.33)	6 (40.00)	4 (33.33)	2 (40.00)	0.976
Shivers	8 (26.67)	7 (46.67)	4 (33.33)	1 (20.00)	0.586
Myalgia	6 (20.00)	5 (33.33)	3 (25.00)	3 (60.00)	0.264
Fever	5 (16.67)	5 (33.33)	1 (8.33)	1 (20.00)	0.379
Arthralgia	4 (13.33)	2 (13.33)	2 (16.67)	1 (20.00)	0.951
Irritability	5 (16.67)	3 (20.00)	1 (8.33)	0 (0.00)	0.819
Application site-pain	1 (3.33)	1 (6.67)	1 (8.33)	1 (20.00)	0.301
Fatigue	1 (3.33)	0 (0.00)	0 (0.00)	0 (0.00)	1.000

## Data Availability

The data that support the findings of this study are available on request from the corresponding author.

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
