# Peer review of "Efficacy and Safety of Heterologous Booster Vaccination after Ad5-nCoV (CanSino Biologics) Vaccine: A Preliminary Descriptive Study"

_vaccines, 2022, doi:10.3390/vaccines10030400_

Round 1

Reviewer 1 Report

The present clinical study has evaluated the effect of heterologous booster vaccination after initial Ad5-nCoV (CanSino Biologics) vaccine. In general, this is a descriptive clinical study with small number of subjects. The data presented Tables 1-4 and Fig. 1 support the stimulation/cross-stimulation of antibody responses by both homologous and heterologous vaccinations. Appropriate control subjects and timing of vaccination were well analyzed in this study that is focused on clinical, demographic and adverse effects of heterologous vaccine boosters.

The authors claim that data in Fig. 1 shows that recipients of the heterologous booster vaccination with either ChAdOx1-S-nCoV-19, Ad26.COV2.S, BNT162b2 or mRNA produced significantly higher levels of antibodies with virus neutralizing capacity than those who only received the Ad5-nCoV vaccine. However,  there is lack of clear difference between the aggregate data in Fig. 1a for homologous booster vaccination compared to aggregated data for heterologous vaccination in Fig. 1b. This point must be clarified in the manuscript that heterologous booster is better than homologous boosting with Ad5-nCoV vaccine after six months.

Author Response

Commentaries: The present clinical study has evaluated the effect of heterologous booster vaccination after initial Ad5-nCoV (CanSino Biologics) vaccine. In general, this is a descriptive clinical study with small number of subjects. The data presented Tables 1-4 and Fig. 1 support the stimulation/cross-stimulation of antibody responses by both homologous and heterologous vaccinations. Appropriate control subjects and timing of vaccination were well analyzed in this study that is focused on clinical, demographic and adverse effects of heterologous vaccine boosters.

The authors claim that data in Fig. 1 shows that recipients of the heterologous booster vaccination with either ChAdOx1-S-nCoV-19, Ad26.COV2.S, BNT162b2 or mRNA produced significantly higher levels of antibodies with virus neutralizing capacity than those who only received the Ad5-nCoV vaccine. However, there is lack of clear difference between the aggregate data in Fig. 1a for homologous booster vaccination compared to aggregated data for heterologous vaccination in Fig. 1b. This point must be clarified in the manuscript that heterologous booster is better than homologous boosting with Ad5-nCoV vaccine after six months.

Answer: We greatly appreciate your comments. Figure 1a compares the % of neutralization between individuals with and without a booster dose, while figure 1b compares only individuals with a booster dose classified according to the vaccine applied. It was specified in the text to avoid confusion (the change was highlighted in green color). We also specify this in the x-axis legends in Figure 1 and the figure footnotes. (lines 128-133)

*The paper has been carefully revised by a professional language editor to improve the grammar and readability.

Reviewer 2 Report

Reviewer’s Comments - Manuscript Vaccines-1588927 “Efficacy and safety of heterologous booster vaccination after Ad5-nCoV (CanSino Biologicalls) vaccine: a preliminary descriptive study” by Josè-Francisco Muñoz-Valle et al.

In the manuscript the Authors evaluate the generation of SARS-Cov-2 neutralizing antibodies 6 months after a single dose of the Ad5-nCoV vaccine (by CanSino Biologics) (62 subjects) and following one heterologous boost (62 subjects) with another vaccine (ChAdOx-1-S-nCoV19. Ad26.COV2.S, BNT162b2 or mRNA-127). The study was done in volunteers in Mexico. The reported results indicate that the prime-boost regimen is safe and induce stronger neutralizing antibodies than one immunization with Ad5-nCoV.   

The study certainly fits the scope of the journal and the topic will be of interest to the vast readership of Vaccines Journal and more specifically to researchers working in the field of Sars-CoV-2 research and vaccine development. However, in my opinion, the Authors may improve the presentation of their results and should clarify few points.

Materials and Methods

-Lines 61-66: please rephrase the entire sentence. In its present form it is difficult to read

-Lines 80-81: when was collected the blood after the boost? Only one time point was collected for both one vaccination and two? Was the test done on serum or plasma? Please clarify in the text.

-Section “Quantification of neutralizing antibodies”: please describe the entire procedure and what variant/s of virus is the kit designed for?

-Lines 100-101: please check if the sentence is complete.

Results/discussion

-Table 1 and 2, please spell at least in the footnotes what SAH (Table 1) and NSAIDs (Table 2) mean.

-Please clarify what are the numbers in parenthesis in the different columns in all tables (1-3)

- It is not reported if anti-Ad antibodies were measured, and if so could you please explain why.

-Were the volunteers samples tested for cross-neutralization to other circulating variants (i.e omicron?)

-Lines 240-241: could you clarify/explain why there is no access to people vaccinated with 2 doses of Ad5-nCoV

-

References

Control ref 3, 4, 6, 12, 19 they seem incomplete to me

Is ref 7 cited at line 39 a proper citation at this point?

I think that in addition to ref 1 cited at line 26 you should cite more adequate ref.

Other points

Please check word spelling in whole manuscript.

In my opinion the manuscript should be revised by a native English speaking

Author Response

Commentaries: In the manuscript the Authors evaluate the generation of SARS-Cov-2 neutralizing antibodies 6 months after a single dose of the Ad5-nCoV vaccine (by CanSino Biologics) (62 subjects) and following one heterologous boost (62 subjects) with another vaccine (ChAdOx-1-S-nCoV19. Ad26.COV2.S, BNT162b2 or mRNA-127). The study was done in volunteers in Mexico. The reported results indicate that the prime-boost regimen is safe and induce stronger neutralizing antibodies than one immunization with Ad5-nCoV.  

The study certainly fits the scope of the journal and the topic will be of interest to the vast readership of Vaccines Journal and more specifically to researchers working in the field of Sars-CoV-2 research and vaccine development. However, in my opinion, the Authors may improve the presentation of their results and should clarify few points.

Materials and Methods

-Lines 61-66: please rephrase the entire sentence. In its present form it is difficult to read

Answer: We appreciate the comment; the correction has been made (lines 64-69)

-Lines 80-81: when was collected the blood after the boost? Only one time point was collected for both one vaccination and two? Was the test done on serum or plasma? Please clarify in the text.

Answer: We clarify this observation in the text (lines 78-80 and 87-92).

-Section “Quantification of neutralizing antibodies”: please describe the entire procedure and what variant/s of virus is the kit designed for?

Answer: Thank you. The entire procedure and what variant of virus the kit was designed for were described (Lines 93-102).

-Lines 100-101: please check if the sentence is complete.

Answer: Thank you for the observation, the correction was made (lines 109-111).

Results/discussion

-Table 1 and 2, please spell at least in the footnotes what SAH (Table 1) and NSAIDs (Table 2) mean.

Answer: Thank you for the observation, the corrections were made (Tables 1-2).

-Please clarify what are the numbers in parenthesis in the different columns in all tables (1-3)

Answer: Thank you for the observation; this information is specified next to the variable (superscript) in column 1 of tables 1-2. For tables 3-4 we specified that the numbers in parenthesis are percentages.

- It is not reported if anti-Ad antibodies were measured, and if so could you please explain why.

Answer: Anti-Ad5 antibodies were not determined in the present study since we had already determined them in the first phase of this study [(Vaccines (Basel). 2021 Sep 20;9(9):1047].

-Were the volunteers samples tested for cross-neutralization to other circulating variants (i.e omicron?)

Answer: The production of antibodies was not determined considering any circulating variants. We will take this suggestion into account for further studies.

-Lines 240-241: could you clarify/explain why there is no access to people vaccinated with 2 doses of Ad5-nCoV

Answer: There is no access to people vaccinated with 2 doses of Ade5-nCoV because the Ade5-nCoV vaccine was designed as a single shot. This information was added in lines 253-255.

References

Control ref 3, 4, 6, 12, 19 they seem incomplete to me

Answer: References were revised and completed

Is ref 7 cited at line 39 a proper citation at this point?

Answer: The citation is correct; however, we added the citations from the 2 described clinical trials.

I think that in addition to ref 1 cited at line 26 you should cite more adequate ref.

Answer: Thanks, we added a new reference that supports this statement.

Other points

Please check word spelling in whole manuscript.

In my opinion the manuscript should be revised by a native English speaking

Answer: Thank you for your commentaries; *The paper has been carefully revised by a professional language editor to improve the grammar and readability.

Reviewer 3 Report

Dear Researcher,

The submitted paper entitled “Efficacy and safety of heterologous booster vaccination after 2
Ad5-nCoV (CanSino Biologics) vaccine: a preliminary descriptive study” aimed to investigate the levels of neutralizing antibodies in subjects exposed to booster dose or not. The study seems to evidence-based research that addresses the myth whether booster with the different covid vaccine is benefited or not.

However, I am confused with the study design. From the write-up it looks that the individuals (professors) were invited for the study over a period of 6 months. Having booster dose of Covid-19 with different vaccines. For parallel comparison, matching of individuals were also “selected” with same age, gender, treatment, COVID-19 history and baseline IgG levels. The “recruited individuals” were also assessed for possible untoward effects. From these information, it looks like a “ clinical Trail” that has to be registered first. Therefore, authors are requested clearly define how the individuals were selected and may be mentioned that authors had no control on the selection of type of vaccine. Reviewer also could not find any information that describe the nature of descriptive study as mentioned in the Title. If the individuals are selected and were matched with non-booster individuals, then may be prospective cohort study?

A few minor changes need to addressed as 

Line 36            It may be written as “….Pakistan and…”

Line 38            Consider revising. “….and good safety and immunogenicity profiles were reported in the first and second phase clinical trials........”

Line 42            Consider revising. Replace “of the health authorities” by “for the health authorities”

Line 46            ….vaccination ;…. may be written as “…..vaccination;…”

Line 47            It is better to mention here the origin/commercial production of this vaccine (Pfizer-BioN-59 Tech). This appears later in the text.

Line 49            “….bases…” may be written as “….basis….”

Line 56-57 Please consider revising “nor the effects of other heterologous booster regimens with different vaccines were evaluated”

Line 71-32 Consider revising the sentence.

Line 77            It is better to mention “N-124” here to avoid confusion

Line 79            Better finish the sentence and start with "The first survey was at the time of.........................”

Line 90            Please mention the Lot/article No. of kit

Line 108, 111. Please delete p>0.05

Table 2. Sd or SD?

Line 199. Sorry, I would recommend not to use phrase : analysis of variance” if authors are comparing two groups. Moreover, kindly clearly mention which variables were tested for Wilcoxin Rank test.

Line 122 (Figure 1 A and B)   In figure 1 a, authors have mentioned the vaccine like Ad5-nCoV (CanSino Biologics). In figure 1 b, the details of manufacturer is mentioned. There should be uniformity in the figures, please. It is better to mention AstraZeneca (ChAdOx1-S-nCoV-19), Pfizer (BNT162b2), Moderna (mRNA-127), Johnson & Johnson (Janssen COVID-19 vaccine)

Line 140 (Table4) Manufacturer and vaccine produced by the particular manufacturer may be mentioned. AstraZeneca (ChAdOx1-S-nCoV-19), Pfizer (BNT162b2), Moderna (mRNA-127), Johnson & Johnson (Janssen COVID-19 vaccine)

Line 159 It may be written as “….it is necessary to wait…..”

Line 217  Authors are recommended to add the data in the main text  (Result Chapter) as it may provide additional information for researchers.

Line 238 Please give space between “Ade5-nCoV.” and “One”

Author Response

Commentaries:

Dear Researcher,

The submitted paper entitled “Efficacy and safety of heterologous booster vaccination after
Ad5-nCoV (CanSino Biologics) vaccine: a preliminary descriptive study” aimed to investigate the levels of neutralizing antibodies in subjects exposed to booster dose or not. The study seems to evidence-based research that addresses the myth whether booster with the different covid vaccine is benefited or not.

However, I am confused with the study design. From the write-up it looks that the individuals (professors) were invited for the study over a period of 6 months. Having booster dose of Covid-19 with different vaccines. For parallel comparison, matching of individuals were also “selected” with same age, gender, treatment, COVID-19 history and baseline IgG levels. The “recruited individuals” were also assessed for possible untoward effects. From these information, it looks like a “ clinical Trail” that has to be registered first. Therefore, authors are requested clearly define how the individuals were selected and may be mentioned that authors had no control on the selection of type of vaccine. Reviewer also could not find any information that describe the nature of descriptive study as mentioned in the Title. If the individuals are selected and were matched with non-booster individuals, then may be prospective cohort study?

Answer: We appreciate your comment, which is very valuable to us. We have modified the description of materials and methods to clear how the patients were selected. We do not consider this study as a clinical trial because we did not manipulate patients into receiving a vaccine or getting a booster. The present study only describes what happened in some individuals immunized with Ad5-nCov who received a booster dose with a different vaccine; that is, describe if there were any adverse effects and whether neutralizing antibodies increased in response to that vaccination. All this, compared to a group of individuals (with similar characteristics) who only received the Ade5-nCoV vaccine without any booster (lines 64-80).

The individuals included in the present study were selected from a cohort that we previously analyzed (6 months before) [Vaccines (Basel).2021 Sep 20;9(9):1047], but in this new study, we only described a group of them with another aim of the previous study.

A few minor changes need to addressed as 

Line 36            It may be written as “….Pakistan and…”

Answer: The change was made (line 36)

Line 38            Consider revising. “….and good safety and immunogenicity profiles were reported in the first and second phase clinical trials........”

Answer: This paragraph was improved (lines 37-39)

Line 42            Consider revising. Replace “of the health authorities” by “for the health authorities”

Answer: Thank you, the change was made (lines 42-43)

Line 46            ….vaccination ;…. may be written as “…..vaccination;…”

Answer: Thank you, the change was made (line 47)

Line 47            It is better to mention here the origin/commercial production of this vaccine (Pfizer-BioN-Tech). This appears later in the text.

Answer: Thank you, the change was made (line 48)

Line 49            “….bases…” may be written as “….basis….”

Answer: Thank you, the change was made (line 50)

Line 56-57 Please consider revising “nor the effects of other heterologous booster regimens with different vaccines were evaluated”

Answer: That paragraph was considered redundant and was therefore deleted.

Line 71-32 Consider revising the sentence.

Answer: The sentence was improved (lines 64-69)

Line 77            It is better to mention “N-124” here to avoid confusion

Answer: Thank you, the change was made (line 75)

Line 79            Better finish the sentence and start with "The first survey was at the time of.........................”

Answer: Thank you, the change was made (line 78-80)

Line 90            Please mention the Lot/article No. of kit

Answer: We add that information (line 90).

Line 108, 111. Please delete p>0.05

Answer: The change has been made.

Table 2. Sd or SD?

Answer: Standar deviation; SD

Line 199. Sorry, I would recommend not to use phrase : analysis of variance” if authors are comparing two groups. Moreover, kindly clearly mention which variables were tested for Wilcoxin Rank test.

Answer: Thanks for your comments. The suggestion was made, and we removed the Wilcoxon test description because it was not applied.

Line 122 (Figure 1 A and B)  In figure 1 a, authors have mentioned the vaccine like Ad5-nCoV (CanSino Biologics). In figure 1 b, the details of manufacturer is mentioned. There should be uniformity in the figures, please. It is better to mention AstraZeneca (ChAdOx1-S-nCoV-19), Pfizer (BNT162b2), Moderna (mRNA-127), Johnson & Johnson (Janssen COVID-19 vaccine).

Answer: Thanks for your comments. The suggestion was made (Figure 1).

Line 140 (Table4) Manufacturer and vaccine produced by the particular manufacturer may be mentioned. AstraZeneca (ChAdOx1-S-nCoV-19), Pfizer (BNT162b2), Moderna (mRNA-127), Johnson & Johnson (Janssen COVID-19 vaccine)

Answer: Thanks for your comments. The suggestion was made (Tables 2 and 4)

Line 159 It may be written as “….it is necessary to wait…..”

Answer: The change has been made (line 175)

Line 217 Authors are recommended to add the data in the main text  (Result Chapter) as it may provide additional information for researchers.

The change has been made (lines 134-135).

Line 238 Please give space between “Ade5-nCoV.” and “One”

Answer: The change was made (line 249, highlighted in gray)

 English language and style are fine/minor spell check required

Answer: *The paper has been carefully revised by a professional language editor to improve the grammar and readability.

Reviewer 4 Report

Estimated Authors of the paper "Efficacy and safety of heterologous booster vaccination after Ad5-nCoV (CanSino Biologics) vaccine: a preliminary descriptive study", first of all thank you for the opportunity to review this very interesting paper.

In this quasi-experimental study, Munoz-Valle et al report on a sample including 64 School Teachers having received a heterologous booster dose for SARS-CoV-2 and 64 having received only the basal schedule.

This study is, from my Point of View, of certain interest for the international readers of Vaccines for several reasons. Firstly, during the first stages of the worldwide campaign against SARS-CoV-2, several different formulates have been employed in order to cope with the availability of mRNA vaccines. According to this report, and consistently with previous reports, we can figure out that booster doses, irrespective of the formulate employed in the last shot, substantially improve neutralizing antibodies even in subjects vaccianted with Ad5-nCoV. This is of certain importance for healthcare providers, and represents a substantial relief for the managing of booster doses, as it simplifies the delivery of further doses.

In my opinion, the present paper may be substantially accepted for publication, but some improvements are required in the discussion section. MOre precisely, the number of recruited individuals is very small, with some subgroups encompassing less than 10 subjects. As a consequence, a more cautionary interpretation of the results is forcibly required: a specific subsection of the discussion should address this specific limit.

Author Response

Commentaries:

Estimated Authors of the paper "Efficacy and safety of heterologous booster vaccination after Ad5-nCoV (CanSino Biologics) vaccine: a preliminary descriptive study", first of all thank you for the opportunity to review this very interesting paper.

In this quasi-experimental study, Munoz-Valle et al report on a sample including 64 School Teachers having received a heterologous booster dose for SARS-CoV-2 and 64 having received only the basal schedule.

This study is, from my Point of View, of certain interest for the international readers of Vaccines for several reasons. Firstly, during the first stages of the worldwide campaign against SARS-CoV-2, several different formulates have been employed in order to cope with the availability of mRNA vaccines. According to this report, and consistently with previous reports, we can figure out that booster doses, irrespective of the formulate employed in the last shot, substantially improve neutralizing antibodies even in subjects vaccianted with Ad5-nCoV. This is of certain importance for healthcare providers, and represents a substantial relief for the managing of booster doses, as it simplifies the delivery of further doses.

In my opinion, the present paper may be substantially accepted for publication, but some improvements are required in the discussion section. MOre precisely, the number of recruited individuals is very small, with some subgroups encompassing less than 10 subjects. As a consequence, a more cautionary interpretation of the results is forcibly required: a specific subsection of the discussion should address this specific limit.

Answer: We appreciate your comment. We are aware that the main limitation of this study was the sample size. However, we were unable to include more participants because, to date, the CanSino (Ade5-nCoV) Vaccine has been approved for a single-dose application, and there is no indication for a booster. Therefore, we only describe the effects of heterologous vaccination in individuals who, voluntarily and independently of us, decided to apply a booster. We added this statement in the discussion section. Moreover, we moderate the conclusions as suggested by the reviewer. Future studies with a larger sample size are essential; however, this study is the first to describe the heterologous effect specifically of Ad5-nCoV vaccine with a booster dose of ChAdOx1-S-nCoV-19, Ad26. COV2.S, BNT162b2, or mRNA-127. Changes regarding this were highlighted in fuchsia color. (Lines 18-20, 233, 248-249, 251-252).

English language and style are fine/minor spell check required

Answer: The paper has been carefully revised by a professional language editor to improve the grammar and readability.

Round 2

Reviewer 3 Report

Dear Authors, Thanks a lot for incorporting the suggestions